# Fermatean Fuzzy Programming with New Score Function: A New Methodology to Multi-Objective Transportation Problems

**M. K. Sharma** [1], **Kamini** [1], **Arvind Dhaka** [2,*], **Amita Nandal** [2], **Hamurabi Gamboa Rosales** [3], **Francisco Eneldo López Monteagudo** [3,*], **Alejandra García Hernández** [3] and **Vinh Truong Hoang** [4]

[1] Department of Mathematics, Chaudhary Charan Singh University, Meerut 250004, Uttar Pradesh, India
[2] Department of Computer and Communication Engineering, Manipal University Jaipur, Jaipur 3303007, India
[3] Unidad Academica de Ingenieria Electrica, Universidad Autonoma de Zacatecas, Jardin Juarez 147, Centro Historico, Zacatecas 98000, Mexico
[4] Faculty of Computer Science, Ho Chi Minh City Open University, Ho Chi Minh 70000, Vietnam
[*] Correspondence: arvind.neomatrix@gmail.com (A.D.); eneldolm@uaz.edu.mx (F.E.L.M.)

**Abstract:** The aim of this work is to establish a new methodology to tackle the multi-objective transportation problems [MOTP] in a Fermatean fuzzy environment that can deal with all the parameters that possess a conflicting nature. In our research work, we developed a new score function in the context of a fermatean nature for converting fuzzy data into crisp data with the help of the Fermatean fuzzy technique. Then, we introduced an algorithm-based methodology, i.e., the Fermatean Fuzzy Programming approach to tackle transportation problems with multi-objectives. The main purpose of this research work is to give an alternate fuzzy programming approach to handle the MOTP. To justify the potential and validity of our work, numerical computations have been carried out using our proposed methodology.

**Keywords:** multi-objective transportation problem [MOTP]; fuzzy programming [FP]; Fermatean fuzzy programming [FTP]; score function; Fermatean fuzzy transportation problem [FFTP]

## 1. Introduction

In the present scenario of highly competitive market dynamics, there is pressure on transportation managers to conduct the smooth transport of goods and services, i.e., transportation problems are concerned with finding a way by which a decision maker can deliver the product from warehouses to a destination at a minimum cost. Transportation models have many applications in supply chain and logistics for reducing costs. In transportation problems, there are mainly three parameters that must be considered to solve the transportation problem, i.e.,

(a) Availability of goods at bases
(b) Demand for products at endpoint
(c) Per unit cost of goods from $i$th base to $j$th point.

To distribute various goods and services from numerous origins to many termini, Hitchcock [1] proposed the transportation problem [TP]. Classical TP is an unusual type of linear programming problem which is more difficult to explain by the simplex method. So, in the literature, many approaches have been developed to find the initial "basic feasible solution" for classical TP which approaches are "Column Minima", "Method "North-West Corner Rule", "Row Minima Method", "Matrix Minima Method" and "Vogel's Method". In real life, there is a need to challenge optimization in the context of various objectives. So, the decision maker wants to handle various objectives, which may be to minimize cost, time, efficiency, less deterioration of a product and less energy consumption, etc. This type of TP is identified as a multi-objective transportation problem [MOTP]. In a MOTP all the objectives are conflicting in nature and with different scales and units for measurement.

For solving such a type of multi-objective transportation problem, the following methods are used:

(a)　Goal Programming,
(b)　Genetic Algorithm,
(c)　Fuzzy Programming,
(d)　Fuzzy goal programming,
(e)　Geometric Programming and other methods also.

Additionally, Lee et al. [2] gave a goal programming method to evaluate the optimal solutions of TP with multi-objectives in multi-dimensional decision-making approaches. In actual life, there are many situations when the available information is not sufficient to judge or formulate the model of the problem. To deal with fuzziness in the real-world era, Zadeh [3] presented the idea of fuzzy set theory. This notion of fuzzy logic is used for the mathematical representation of less knowledgeable or imprecise data by a membership function. Bellman and Zadeh [4] introduced decision-making problems in such an environment in which goals or constraints are not defined precisely. Oheigeartaigh [5] described transportation problems in real-life situations and developed an algorithm for transporting goods from supply nodes to demand nodes in a fuzzy environment. Zimmermann [6] presented that the results attained by fuzzy linear programming are continuously ideal and effectual. When the "cost factors (C)", "supply(S)" and "demand (D)" measures are identified exactly, many procedures have been established for explaining the TP. However, in the current circumstances, there are numerous cases in which the C, S and D measures are fuzzy amounts. A fuzzy Transportation Problem is a Transportation Problem in which the parameters such as Cost, Supply and Demand measures are in the form of fuzzy numbers. Chanas et al. [7] analyzed the Transportation Problem in fuzzy uncertainty and used a parametric technique to find the optimal solution for Transportation Problems. Chanas and Kuchata [8] proposed a procedure for attaining an optimal solution for the TP with fuzzy parameters expressed in terms of fuzzy numbers. Atanassov [9] initiated the notion of Intuitionistic Fuzzy Set (IFS), a generalization of fuzzy sets that incorporates both truth and false grades with a hesitation margin, such that the sum of the degree of truth and false grades is less than or equal to 1. Then Yager [10] introduced the ortho-pair of fuzzy set [FS], where the square of the sum of truth and a false value is less than equal to 1, called the Pythagorean fuzzy set [PFS]. Senapati and Yager [11] presented the notion of Fermatean Fuzzy Sets [FFSs] to deal with the situation where the fuzzy logic fails due to membership grades only. Several researchers developed the model of the TP by either an individual objective TP or a MOTP, considering various fuzzy contexts. In the case of multi-objective transportation problems, Ringuest [12] proposed a collaborative approach for explaining MOTP and attained more than k-dominated and non-dominated solutions. Bit et al. [13] established a new method for MOTP with the help of fuzzy programming using a linear membership function. The weight and priorities of the objectives are involved in the method. Li and Lai [14] introduced a vague compromise programming method to the MOTP. Then, Ammar [15] characterized the optimal solution based on the alpha level of fuzzy numbers and checked the solidity of MOTP with a fuzzy environment. Lau et al. [16] developed a model by using a genetic algorithm to obtain the solution of MOTP. Kocken et al. [17] established a model by using a compensatory fuzzy technique for multi-objective linear transportation problems with triangular fuzzy numbers of parameters. In this technique, he used Zimmermann's "min" operator and fixed the cost-satisfaction intervals and breaking points. Nomani [18] established a new weight approach based on goal programming for MOTP. Ahmad and Adhami [19] considered nonlinearity in MOTP in the neutrosophic situation. A MOTP with a p-facility localization issue was given by Das and Roy [20]. To establish a real transport network, Das et al. [21] incorporated two-fold ambiguity. Ghosh et al. [22] developed a solid MOTP in intuitionistic circumstances. Then, in TP, Ghosh and Roy et al. [23] observed an extra cost which is considered a type-II fixed charge. Midya et al. [24] developed a solid MOTP with a fixed charge in an intuitionistic fuzzy environment. Sahoo [25] originally solved TP in the Fermatean fuzzy situation

and planned a procedure in which the Fermatean fuzzy transportation problem [FFTP] is transformed into a conventional TP to obtain the best optimal solution. Thereafter, Sahoo [26] anticipated various score functions for converting the Fermatean fuzzy data into a crisp form and applied the TOPOSIS technique to solve the MOTP in the Fermatean fuzzy context. Ghosh [27] introduced the latest technology (Preservation technology) for MOTP with preservation cost for the reduction in the rate of deterioration and also for building a more realistic problem; they considered criteria such that all the parameters of the problem are Pythagorean fuzzy sets; Rani and Ebrahimnejad [28] established a new algorithm to tackle the unbalanced fully rough and fixed charge MOTP.

The basic purposes of this work are as follows:

1. To develop a novel score function in order to rank the fuzzy numbers.
2. To use the credibility of the Fermatean fuzzy numbers to design Fermatean fuzzy programming.
3. To develop a mathematical model based on the Fermatean fuzzy programming for MOTP.
4. To illustrate and validate the proposed approach, we will apply our technique to numeric data.

The work conducted in the research paper is categorized into eight segments including the present one dealing with the introduction and review of the work conducted by many authors. The Section 2 of the manuscript includes a basic definition related to the work in research. In the Section 3, we formulated a mathematical model for MOTP in the Fermatean fuzzy situation. Additionally, in the Section 4, we proposed a novel score function for the ranking of the Fermatean fuzzy numbers. The Section 5 includes the development of the model of the proposed Fermatean Fuzzy Programming for MOTP. In the Section 6, we describe the proposed methodology of this in the context of the proposed score function. The Section 7 includes a mathematical illustration to check the effectiveness of our planned methodology. In the last section, we discuss the conclusion of our planned matter.

## 2. Basic Definitions

### 2.1. Fermatean Fuzzy Set [FFS]

An FFS can be given as:

$$\widetilde{f} = \left\{ \langle w, \theta_{\widetilde{f}}(w), \delta_{\widetilde{f}}(w) : w \in X \rangle \right\}$$

where $\Sigma$ is a universal set and $\theta_{\widetilde{f}}(w) : X \to [0,1]$ is the degree of satisfaction and $\delta_{\widetilde{f}}(w) : X \to [0,1]$ is the degree of dissatisfaction $w \in X$ and these two are related by the relation;

$$0 \leq \left( \theta_{\widetilde{f}}(w) \right)^3 + \left( \delta_{\widetilde{f}}(w) \right)^3 \leq 1 \ \forall \ w \in X.$$

And $\sigma_{\widetilde{f}}(w)$ denotes the grade of indeterminacy of $w \in X$, such that:

$$\sigma_{\widetilde{f}}(w) = \sqrt[3]{1 - \left( \theta_{\widetilde{f}}(w) \right)^3 - \left( \delta_{\widetilde{f}}(w) \right)^3}$$

The set $\widetilde{f} = \left\{ \langle w, \theta_{\widetilde{f}}(w), \delta_{\widetilde{f}}(w) : w \in X \rangle \right\}$ is denoted as $\widetilde{f} = \langle \theta_{\widetilde{f}}, \delta_{\widetilde{f}} \rangle$.

### 2.2. Arithmetic Operation on Fermatean Fuzzy Sets

Let $\widetilde{f} = \langle \theta_{\widetilde{f}}, \delta_{\widetilde{f}} \rangle$, $\widetilde{f_1} = \langle \theta_{\widetilde{f_1}}, \delta_{\widetilde{f_1}} \rangle$ and $\widetilde{f_2} = \langle \theta_{\widetilde{f_2}}, \delta_{\widetilde{f_2}} \rangle$ be three Fermatean fuzzy sets on a universal set $X$ and $\Lambda > 0$ be any scalar then arithmetic operations on the FFSs are defined such that:

(i). $\widetilde{f_1} \oplus \widetilde{f_2} = \langle \theta_{\widetilde{f_1}}, \delta_{\widetilde{f_1}} \rangle \oplus \langle \theta_{\widetilde{f_2}}, \delta_{\widetilde{f_2}} \rangle$

(ii). $= < \sqrt[3]{\left( \theta_{\widetilde{f_1}} \right)^3 + \left( \theta_{\widetilde{f_2}} \right)^3 - \left( \theta_{\widetilde{f_1}} \right)^3 \left( \theta_{\widetilde{f_2}} \right)^3}, \delta_{\widetilde{f_1}} \delta_{\widetilde{f_2}} >$

(iii). $\widetilde{f_1} \otimes \widetilde{f_2} = \langle \theta_{\widetilde{f_1}}, \delta_{\widetilde{f_1}} \rangle \otimes \langle \theta_{\widetilde{f_2}}, \delta_{\widetilde{f_2}} \rangle$

(iv). $= < \theta_{\widetilde{f_1}} \theta_{\widetilde{f_2}}, \sqrt[3]{\left(\delta_{\widetilde{f_1}}\right)^3 + \left(\delta_{\widetilde{f_2}}\right)^3 - \left(\delta_{\widetilde{f_1}}\right)^3 \left(\delta_{\widetilde{f_2}}\right)^3} >$

(v). $\Lambda \odot \widetilde{f} = \langle \sqrt[3]{1 - (1 - (\theta_{\widetilde{f}})^3)^{\Lambda}}, (\delta_{\widetilde{f}})^{\Lambda} \rangle$

(vi). $\widetilde{f}^{\Lambda} = \langle (\theta_{\widetilde{f}})^{\Lambda}, \sqrt[3]{1 - (1 - (\delta_{\widetilde{f}})^3)^{\Lambda}} \rangle$

(vii). $\widetilde{f_1} \cup \widetilde{f_2} = \langle \max\left(\theta_{\widetilde{f_1}}, \theta_{\widetilde{f_2}}\right), \min\left(\delta_{\widetilde{f_1}}, \delta_{\widetilde{f_2}}\right) \rangle$

(viii). $\widetilde{f_1} \cap \widetilde{f_2} = \langle \min\left(\theta_{\widetilde{f_1}}, \theta_{\widetilde{f_2}}\right), \max(\delta_{\widetilde{f_1}}, \delta_{\widetilde{f_2}}) \rangle$

(ix). $\widetilde{f}^c = \delta_{\widetilde{f}}, \theta_{\widetilde{f}}.$

Example

Let $\widetilde{f} = \langle 0.5,\ 0.6 \rangle, \widetilde{f_1} = \langle 0.2,\ 0.8 \rangle, \widetilde{f_2} = \langle 0.9,\ 0.4 \rangle$ be three FFSs and $\Lambda = 3$ be a scalar, then:

(i). $\widetilde{f_1} \oplus \widetilde{f_2} = \langle 0.2,\ 0.8 \rangle \oplus \langle 0.9,\ 0.4 \rangle = \langle 0.9009,\ 0.32 \rangle$

(ii). $\widetilde{f_1} \otimes \widetilde{f_2} = \langle 0.2,\ 0.8 \rangle \otimes \langle 0.9,\ 0.4 \rangle = \langle 0.18,\ 0.8159 \rangle$

(iii). $\Lambda \odot \widetilde{f} = 3 \odot \langle 0.5,\ 0.6 \rangle = \langle 0.6911,\ 0.216 \rangle$

(iv). $\widetilde{f}^{\Lambda} = \langle 0.5,\ 0.6 \rangle^3 = \langle 0.125,\ 0.8032 \rangle$

(v). $\widetilde{f_1} \cup \widetilde{f_2} = \langle 0.2,\ 0.8 \rangle \cup \langle 0.9,\ 0.4 \rangle = \langle 0.9,\ 0.4 \rangle$

(vi). $\widetilde{f_1} \cap \widetilde{f_2} = \langle 0.2,\ 0.8 \rangle \cap \langle 0.9,\ 0.4 \rangle = \langle 0.2,\ 0.8 \rangle$

(vii). $\widetilde{f}^c = \langle 0.5,\ 0.6 \rangle^c = \langle 0.6,\ 0.5 \rangle.$

*2.3. Score Function*

Let $\widetilde{f}$ be a Fermatean fuzzy set $\widetilde{f} = \langle \phi_{\widetilde{F}}, \psi_{\widetilde{F}} \rangle$, then the score function of $\widetilde{f}$ which is denoted by $S_f(\widetilde{f})$ and defined as follows:

$$S_f(\widetilde{f}) = \left( \phi_{\widetilde{F}}{}^3 - \psi_{\widetilde{F}}{}^3 \right)$$

Property: Consider an FFS $\widetilde{F} = \langle \phi_{\widetilde{F}}, \psi_{\widetilde{F}} \rangle$, then $S_F^*(\widetilde{F}) \in [0, 1]$.

**Proof.** By the definition of an ortho pair, $\phi_{\widetilde{F}}, \psi_{\widetilde{F}} \in [0, 1]$. Then, $\min\left(\phi_{\widetilde{F}}, \psi_{\widetilde{F}}\right) \in [0, 1]$. □

Also, $\phi_{\widetilde{F}}{}^3 \geq 0,\ \psi_{\widetilde{F}}{}^3 \geq 0,\ \phi_{\widetilde{F}}{}^3 \leq 1$ and $\psi_{\widetilde{F}}{}^3 \leq 1$

$\implies 1 - \psi_{\widetilde{F}}{}^3 \geq 0$

$\implies 1 + \phi_{\widetilde{F}}{}^3 - \psi_{\widetilde{F}}{}^3 \geq 0$

$\therefore \frac{1}{2}\left(1 + \phi_{\widetilde{F}}{}^3 - \psi_{\widetilde{F}}{}^3\right) \cdot \left(\min(\phi_{\widetilde{F}}, \psi_{\widetilde{F}})\right) \geq 0$

Again, $\phi_{\widetilde{F}}{}^3 - \psi_{\widetilde{F}}{}^3 \leq 1$

$\implies 1 + \phi_{\widetilde{F}}{}^3 - \psi_{\widetilde{F}}{}^3 \leq 2\ (\because \phi_{\widetilde{F}}{}^3 \geq 0)$

$\implies \frac{1 + \phi_{\widetilde{F}}{}^3 - \psi_{\widetilde{F}}{}^3}{2} \leq 1$

$\implies \frac{1}{2}\left(1 + \phi_{\widetilde{F}}{}^3 - \psi_{\widetilde{F}}{}^3\right) \cdot \left(\min(\phi_{\widetilde{F}}, \psi_{\widetilde{F}})\right) \cdot 1\ (\because \min(\phi_{\widetilde{F}}, \psi_{\widetilde{F}}) \leq 1)$

Hence, $S_F^*(\widetilde{F}) \in [0, 1]$.

*2.4. Accuracy Function*

Let $\widetilde{f}$ be a Fermatean fuzzy set $\widetilde{f} = \langle \theta_{\widetilde{f}}, \delta_{\widetilde{f}} \rangle$, then the accuracy function of $\widetilde{f}$ which is denoted by $\hat{A}_f(\widetilde{f})$ and defined as follows:

$$\hat{A}_f(\widetilde{f}) = \left( \theta_{\widetilde{f}}{}^3 + \delta_{\widetilde{f}}{}^3 \right)$$

## 3. Mathematical Formulation for Multi-Objective Transportation Problem in a Fermatean Fuzzy Environment

Let us consider a MOTP with $k$ objectives containing $m$ supply nodes and $n$ demand nodes. Additionally, $a_i^{\widetilde{f}} = \langle \theta_{\widetilde{a_i}}, \delta_{\widetilde{a_i}} \rangle$ units are available at the $i^{th}$ supply node and $b_j^{\widetilde{f}} = \langle \theta_{\widetilde{b_j}}, \delta_{\widetilde{b_j}} \rangle$

units are demanded on $j^{th}$ demand node. Suppose $\widetilde{c_{ij}^{f}} = \langle \theta_{\widetilde{c}_{ij}}, \delta_{\widetilde{c}_{ij}} \rangle$ is the unit fermatean fuzzy transportation cost and the $i^{th}$ source node to $j^{th}$ demand node and $\sigma_{ij}$ is the number of items that are carried from $i^{th}$ source node to $j^{th}$ demand node.

The mathematical formulation for the MOTP in the Fermatean Fuzzy Situation is as follows:

$$\min F_K = \sum_{i=1}^{m} \sum_{j=1}^{n} \widetilde{c_{ijK}^{f}} w_{ij} \, , \, K = 1, \, 2, \, 3, \, 4, \ldots, \, k$$

$$\text{s.t.} \sum_{j=1}^{n} w_{ij} \leq \widetilde{a_i^f}, \, i = 1, \, 2, \, 3, \, \ldots, \, m$$

$$\sum_{i=1}^{m} w_{ij} \geq \widetilde{b_j^f}, \, j = 1, \, 2, \, 3, \, \ldots, \, n$$

Such that $\widetilde{a_i^f} = \langle \theta_{\widetilde{a}_i}, \delta_{\widetilde{a}_i} \rangle$ where $0 \leq \left( \theta_{\widetilde{a}_i} \right)^3 + \left( \delta_{\widetilde{a}_i} \right)^3 \leq 1,$

$\widetilde{b_j^f} = \langle \theta_{\widetilde{b}_j}, \delta_{\widetilde{b}_j} \rangle$ where $0 \leq \left( \theta_{\widetilde{b}_j} \right)^3 + \left( \delta_{\widetilde{b}_j} \right)^3 \leq 1$

$\widetilde{c_{ij}^f} = \langle \theta_{\widetilde{c}_{ij}}, \delta_{\widetilde{c}_{ij}} \rangle$ where $0 \leq \left( \theta_{\widetilde{c}_{ij}} \right)^3 + \left( \delta_{\widetilde{c}_{ij}} \right)^3 \leq 1 w_{ij} \geq 0, \forall \, i, j$

## 4. Proposed Score Function for Fermatean Fuzzy Sets

### 4.1. Fermatean Fuzzy Score Functions

Let $\widetilde{f} = \langle \theta_{\widetilde{f}}, \delta_{\widetilde{f}} \rangle$ be any Fermatean fuzzy number and then the score function of $\widetilde{f}$ which is denoted by $S\left(\widetilde{f}\right)$, defined as follows:

$$S\left(\widetilde{f}\right) = \frac{1}{2}\left(1 + \theta_{\widetilde{f}} - \delta_{\widetilde{f}}\right)\left(\min(\theta_{\widetilde{f}}, \delta_{\widetilde{f}})\right)^2$$

### 4.2. Property

Let $\widetilde{f} = \langle \theta_{\widetilde{f}}, \delta_{\widetilde{f}} \rangle$ be any Fermatean fuzzy set, then the score function of $\widetilde{f}$, $S\left(\widetilde{f}\right) \in [0, 1]$.

**Proof.** By the description of membership as well as non-membership pairs, $\theta_{\widetilde{f}}, \delta_{\widetilde{f}} \in [0, 1]$. $\square$

Then, $\min\left(\theta_{\widetilde{f}}, \delta_{\widetilde{f}}\right) \in [0, 1]$.

Additionally, $\theta_{\widetilde{f}} \geq 0, \delta_{\widetilde{f}} \geq 0$ , $\theta_{\widetilde{f}} \leq 1$ and $\delta_{\widetilde{f}} \leq 1$

$$\implies 1 - \delta_{\widetilde{f}} \geq 0 \implies 1 + \theta_{\widetilde{f}} - \delta_{\widetilde{f}} \geq 0$$

$\therefore \frac{1}{2}\left(1 + \theta_{\widetilde{f}} - \delta_{\widetilde{f}}\right)\left(\min\left(\theta_{\widetilde{f}}, \delta_{\widetilde{f}}\right)\right)^2 \geq 0,$

Hence, $S\left(\widetilde{f}\right) \geq 0$.

Again, $\theta_{\widetilde{f}} \leq 1$ and $\delta_{\widetilde{f}} \leq 1 \implies \theta_{\widetilde{f}} - \delta_{\widetilde{f}} \leq 1$

$$\implies 1 + \theta_{\widetilde{f}} - \delta_{\widetilde{f}} \leq 1 + 1 = 2 \implies \frac{1 + \theta_{\widetilde{f}} - \delta_{\widetilde{f}}}{2} \leq 1$$

and $\min\left(\theta_{\widetilde{f}}, \delta_{\widetilde{f}}\right) \leq 1 \implies \left(\min(\theta_{\widetilde{f}}, \delta_{\widetilde{f}})\right)^2 \leq 1, \implies \frac{1}{2}\left(1 + \theta_{\widetilde{f}} - \delta_{\widetilde{f}}\right)\left(\min(\theta_{\widetilde{f}}, \delta_{\widetilde{f}})\right)^2 \leq 1,$

Hence, $S\left(\widetilde{f}\right) \leq 1 \implies S\left(\widetilde{f}\right) \in [0, 1]$.

## 5. Proposed Model for Fermatean Fuzzy Programming

Senapati and Yager [11] introduced the extension of an intuitionistic and Pythagorean fuzzy set when the sum of truth and false grades with the sum of the square of truth grade and false grade is greater than 1, but the sum of the cube of truth grade and false grade is less than equal to the 1. These fuzzy sets are FFS. FFS are more realistic and handle more uncertainty than Intuitionistic and Pythagorean fuzzy sets. Due to such an environment, Zimmermann [6] introduced a fuzzy programming approach for multi-objective decision-making problems based on a min-max operator. In this approach, we can use linear, exponential, or hyperbolic truth functions to attain compromised optimal solutions to the problems. Then, intuitionistic fuzzy programming is also developd for multi-objective problems in an intuitionistic fuzzy environment in which the truth and false grades may be linear, exponential, or hyperbolic functions. After that, Pythagorean fuzzy programming is also developed for such problems in a fuzzy environment. Now, we introduce non-linear programming, Fermatean Fuzzy Programming, to obtain a compromise optimal solution to

all objectives simultaneously of multi-objective decision-making problems in a Fermatean fuzzy environment and in any other environment, defined as:

Let $U_k$ and $L_k$ be upper and lower bounds, respectively, for objective $F_k(w)$ of the problem and $\mu(F_k(w))$ be membership function for objective $F_k(w)$ and $\vartheta(F_2(w))$ be non-membership function for the objective function $F_k(w)$. Then, the proposed model for Fermatean Fuzzy Programming is as follows:

$$\text{Max. } \sigma\, \gamma_1{}^3 - \gamma_2{}^3$$

where

$$\mu(F_k(w))^{\,3} \geq \gamma_1{}^3,\ \forall\, k \qquad \vartheta(F_k(w))^{\,3} \leq \gamma_2{}^3,\ \forall\, k$$

where

$$\mu(F_k(w)) = \begin{cases} 1, & if\ F_k(w)\ \leq L_k \\ \frac{U_k - F_k(w)}{U_k - L_k}, & if\ L_k \leq F_k(\sigma) \leq U_k \\ 0, & if\ F_k(w) \geq U_k \end{cases}$$

and

$$\vartheta(F_k(w)) = \begin{cases} 0, & if\ F_k(w)\ \leq L_k \\ \frac{F_k(w) - L_k}{U_k - L_k}, & if\ L_k \leq F_k(\sigma) \leq U_k \\ 1, & if\ F_k(w) \geq U_k \end{cases}$$

i.e., $\left(U_k - F_k(w)\right)^3 \geq d_k{}^3\gamma_1{}^3$

$\left(F_k(w) - L_k\right)^3 \leq d_k{}^3\gamma_2{}^3$, where $d_k = U_k - L_k$.

with respect to the constraints,

$$w_{11} + w_{12} + \ldots\ldots\ldots + w_{1n} \leq a_1$$
$$w_{21} + w_{22} + \ldots\ldots\ldots + w_{2n} \leq a_2$$
$$.$$
$$.$$
$$w_{m1} + w_{m2} + \ldots\ldots\ldots + w_{mn} \leq a_m$$
$$w_{11} + w_{21} + \ldots\ldots\ldots + w_{m1} \geq b_1,$$
$$w_{12} + w_{22} + \ldots\ldots\ldots + w_{m2} \geq b_2,$$
$$.$$
$$.$$
$$w_{1n} + w_{2n} + \ldots\ldots\ldots + w_{mn} \geq b_n$$

and $\sum\limits_{i=1}^{m} a_i = \sum\limits_{j=1}^{n} b_j$, $w_{ij} \geq 0$, $0 \leq \gamma_1{}^3$, $\gamma_2{}^3 \leq 1$, $0 \leq \gamma_1{}^3 + \gamma_2{}^3 \leq 1$ and $\gamma_1{}^3 \geq \gamma_2{}^3$.

## 6. Proposed Methodology

To handle the MOTP in the Fermatean fuzzy environment, we propose a methodology. The steps involved in the proposed methodology are depicted as follows:

Step 1: First, consider a MOTP in Fermatean fuzzy uncertainty such as:

$$\min F_K = \sum_{i=1}^{m} \sum_{j=1}^{n} \tilde{c}_{ijK}^{f} w_{ij}\ ,\ K = 1,\,2,\,3,\,4,\ldots,\,k$$
$$\text{s.t. } \sum_{j=1}^{n} w_{ij} \leq \tilde{a}_i^{f},\ i = 1,\,2,\,3,\,\ldots,\,m$$
$$\sum_{i=1}^{m} w_{ij} \geq \tilde{b}_j^{f},\ j = 1,\,2,\,3,\,\ldots,\,n$$

Such that $\tilde{a}_i^{f} = \langle \theta_{\tilde{a}_i}, \delta_{\tilde{a}_i} \rangle$ where $0\ \leq \left(\theta_{\tilde{a}_i}\right)^3 + \left(\delta_{\tilde{a}_i}\right)^3 \leq 1,$

$\tilde{b}_j^{f} = \langle \theta_{\tilde{b}_j}, \delta_{\tilde{b}_j} \rangle$ where $0\ \leq \left(\theta_{\tilde{b}_j}\right)^3 + \left(\delta_{\tilde{b}_j}\right)^3 \leq 1,$

$\tilde{c}_{ij}^{f} = \langle \theta_{\tilde{c}_{ij}}, \delta_{\tilde{c}_{ij}} \rangle$ where $0\ \leq \left(\theta_{\tilde{c}_{ij}}\right)^3 + \left(\delta_{\tilde{c}_{ij}}\right)^3 \leq 1,$

$w_{ij} \geq 0, \forall\, i, j$

Step 2: Then, convert the Fermatean fuzzy data into crisp data by using the proposed score function for Fermatean fuzzy sets as:

$$\min F_K = \sum_{i=1}^{m} \sum_{j=1}^{n} S(\tilde{c}_{ijK}^{f}) w_{ij} , \; K = 1, 2, 3, 4, \ldots, k$$

$$\text{s.t.} \sum_{j=1}^{n} w_{ij} \leq \; S(\tilde{a}_{i}^{f}), \; i = 1, 2, 3, \ldots, m$$

$$\sum_{i=1}^{m} w_{ij} \geq \; S\left(\tilde{b}_{j}^{f}\right), \; j = 1, 2, 3, \ldots, n$$

Step 3: Now, solve this problem for all objectives by taking one objective at a time. We obtain basic feasible solutions for all objectives.

Step 4: Now, we built a pay-off matrix for all objectives then we obtain upper bound $U_k$ and lower bound $L_k$ for the objective $F_k(w)$ through pay-off matriσ.

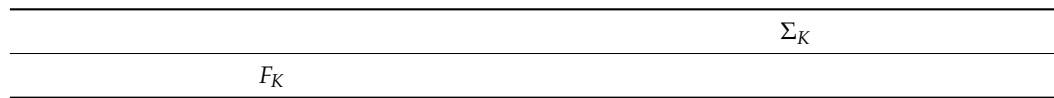

Step 5: Then, built a model of the problem by proposed Fermatean fuzzy programming approach and then solve this model by Lingo 19.0 software. The architecture is shown in Figure 1. The numerical computations of the proposed technique are give in Tables 1–3.

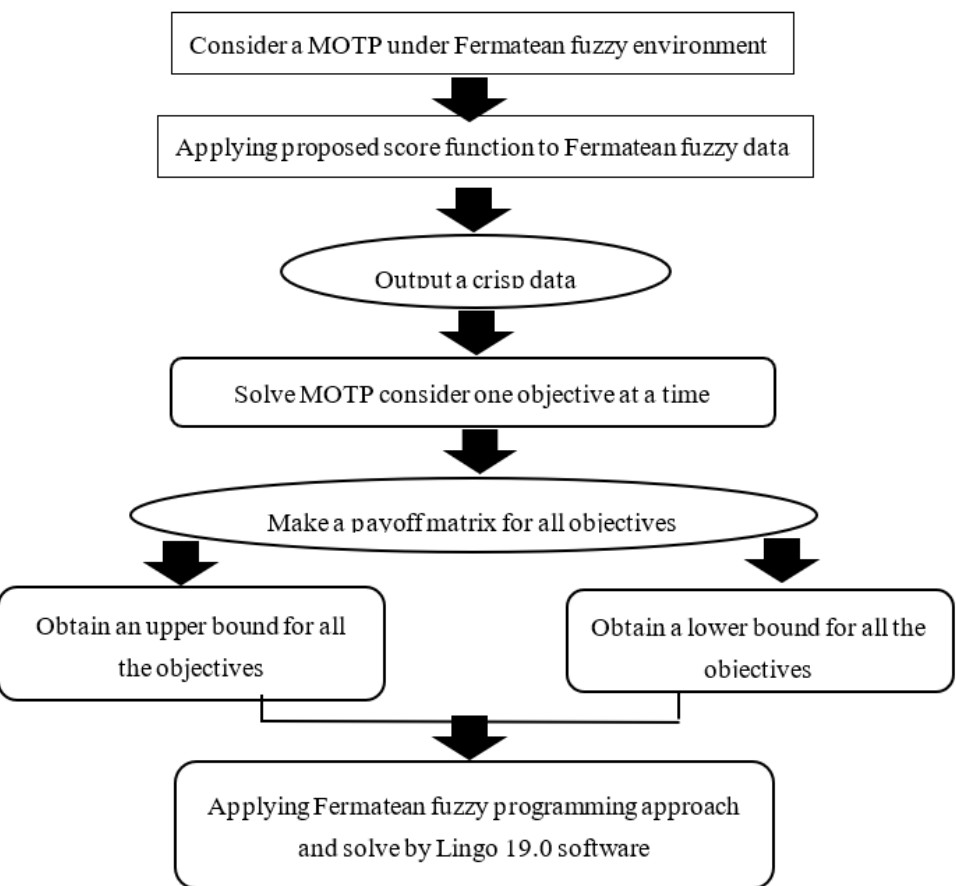

**Figure 1.** Structure outlines of proposed algorithm.

## 7. Numerical Computations

We consider a MOTP in Fermatean fuzzy uncertainty where all the variables of the problem are Fermatean fuzzy numbers. This is as follows:

**Table 1.** First Objective Cost.

|  | $d_1$ | $d_2$ | $d_3$ | $d_4$ | Supply |
|---|---|---|---|---|---|
| $o_1$ | (0.2, 0.5) | (0.2, 0.4) | (0.3, 0.7) | (0.6, 0.5) | (0.6, 0.4) |
| $o_2$ | (0.3, 0.5) | (0.2, 0.9) | (0.7, 0.1) | (0.4, 0.7) | (0.3, 0.5) |
| $o_3$ | (0.7, 0.1) | (0.2, 0.3) | (0.5, 0.1) | (0.6, 0.4) | (0.4, 0.8) |
| Dd | (0.2, 0.5) | (0.4, 0.7) | (0.6, 0.4) | (0.2, 0.5) | - |

**Table 2.** Second Objective Cost.

|  | $d_1$ | $d_2$ | $d_3$ | $d_4$ | Supply |
|---|---|---|---|---|---|
| $o_1$ | (0.4, 0.7) | (0.3, 0.7) | (0.6, 0.5) | (0.2, 0.8) | (0.6, 0.4) |
| $o_2$ | (0.3, 0.5) | (0.4, 0.8) | (0.3, 0.7) | (0.6, 0.2) | (0.3, 0.5) |
| $o_3$ | (0.4, 0.7) | (0.4, 0.6) | (0.3, 0.1) | (0.5, 0.1) | (0.4, 0.8) |
|  | (0.2, 0.5) | (0.4, 0.7) | (0.6, 0.4) | (0.2, 0.5) |  |

**Table 3.** Third Objective Cost.

|  | $d_1$ | $d_2$ | $d_3$ | $d_4$ | Supply |
|---|---|---|---|---|---|
| $o_1$ | (0.2, 0.3) | (0.6, 0.7) | (0.3, 0.7) | (0.6, 0.5) | (0.6, 0.4) |
| $o_2$ | (0.7, 0.1) | (0.2, 0.9) | (0.6, 0.7) | (0.4, 0.7) | (0.3, 0.5) |
| $o_3$ | (0.8, 0.4) | (0.3, 0.7) | (0.2, 0.5) | (0.7, 0.1) | (0.4, 0.8) |
|  | (0.2, 0.5) | (0.4, 0.7) | (0.6, 0.4) | (0.2, 0.5) |  |

Step 2: Now, Convert Fermatean fuzzy parameters into crisp form by applying the proposed score function as:

Supply:

$$
\begin{aligned}
S(a_{\widetilde{f_1}}) = S(\langle 0.6, 0.4 \rangle) &= \tfrac{1}{2}(1 + 0.6 - 0.4)(\min(0.6,\ 0.4))^2 \\
&= \tfrac{1}{2}(1.2)(0.4)^2 = 0.6 \times 0.16 = 0.096 \\
S(a_{\widetilde{f_2}}) = S(\langle 0.3, 0.5 \rangle) &= \tfrac{1}{2}(1 + 0.3 - 0.5)(\min(0.3,\ 0.5))^2 \\
&= \tfrac{1}{2}(0.8)(0.3)^2 = 0.4 \times 0.09 = 0.036 \\
S(a_{\widetilde{f_3}}) = S(\langle 0.4, 0.8 \rangle) &= \tfrac{1}{2}(1 + 0.4 - 0.8)(\min(0.4,\ 0.8))^2 \\
&= \tfrac{1}{2}(0.6)(0.4)^2 = 0.3 \times 0.16 = 0.048
\end{aligned}
$$

Demand:

$$
\begin{aligned}
S(b_{\widetilde{f_1}}) = S(\langle 0.2, 0.5 \rangle) &= \tfrac{1}{2}(1 + 0.2 - 0.5)(\min(0.2,\ 0.5))^2 \\
&= \tfrac{1}{2}(0.7)(0.2)^2 = 0.35 \times 0.04 = 0.014 \\
S(b_{\widetilde{f_2}}) = S(\langle 0.4, 0.7 \rangle) &= \tfrac{1}{2}(1 + 0.4 - 0.7)(\min(0.4,\ 0.7))^2 \\
&= \tfrac{1}{2}(0.7)(0.4)^2 = 0.35 \times 0.16 = 0.056 \\
S(b_{\widetilde{f_3}}) = S(\langle 0.6, 0.4 \rangle) &= \tfrac{1}{2}(1 + 0.6 - 0.4)(\min(0.6,\ 0.4))^2 \\
&= \tfrac{1}{2}(1.2)(0.4)^2 = 0.6 \times 0.16 = 0.096 \\
S(b_{\widetilde{f_4}}) = S(\langle 0.2, 0.5 \rangle) &= \tfrac{1}{2}(1 + 0.2 - 0.5)(\min(0.2,\ 0.5))^2 \\
&= \tfrac{1}{2}(0.7)(0.2)^2 = 0.35 \times 0.04 = 0.014
\end{aligned}
$$

Since $\sum\limits_{i=1}^{m} S(a_{\widetilde{f_i}}) = \sum\limits_{j=1}^{n} S(b_{\widetilde{f_j}})$, then the problem is a balanced multi-objective transportation problem.

Costs:

First objective costs:

$$S(c_{\widetilde{f}_{11}}) = S(\langle 0.2, \ 0.5 \rangle) \quad = \tfrac{1}{2}(1 + 0.2 - 0.5)(\min(0.2, \ 0.5))^2$$
$$= \tfrac{1}{2}(0.7)(0.2)^2 = 0.35 \times 0.04 = 0.014$$
$$S(c_{\widetilde{f}_{12}}) = S(\langle 0.2, \ 0.4 \rangle) \quad = \tfrac{1}{2}(1 + 0.2 - 0.4)(\min(0.4, \ 0.2))^2$$
$$= \tfrac{1}{2}(0.8)(0.2)^2 = 0.4 \times 0.04 = 0.016$$
$$S(c_{\widetilde{f}_{13}}) = S(\langle 0.3, \ 0.7 \rangle) = \tfrac{1}{2}(1 + 0.3 - 0.7)(\min(0.3, \ 0.7))^2$$
$$= \tfrac{1}{2}(0.6)(0.3)^2 = 0.3 \times 0.09 = 0.027$$
$$S(c_{\widetilde{f}_{14}}) = S(\langle 0.6, \ 0.5 \rangle) \quad = \tfrac{1}{2}(1 + 0.6 - 0.5)(\min(0.6, \ 0.5))^2$$
$$= \tfrac{1}{2}(1.1)(0.5)^2 = 0.55 \times 0.25 = 0.1375$$
$$S(c_{\widetilde{f}_{21}}) = S(\langle 0.3, \ 0.5 \rangle) \quad = \tfrac{1}{2}(1 + 0.3 - 0.5)(\min(0.3, \ 0.5))^2$$
$$= \tfrac{1}{2}(0.8)(0.3)^2 = 0.4 \times 0.09 = 0.036$$
$$S(c_{\widetilde{f}_{22}}) = S(\langle 0.2, \ 0.9 \rangle) \quad = \tfrac{1}{2}(1 + 0.2 - 0.9)(\min(0.2, \ 0.9))^2$$
$$= \tfrac{1}{2}(0.3)(0.2)^2 = 0.15 \times 0.04 = 0.006$$
$$S(c_{\widetilde{f}_{23}}) = S(\langle 0.7, \ 0.1 \rangle) \quad = \tfrac{1}{2}(1 + 0.7 - 0.1)(\min(0.7, \ 0.1))^2$$
$$= \tfrac{1}{2}(1.6)(0.1)^2 = 0.8 \times 0.01 = 0.008$$
$$S(c_{\widetilde{f}_{24}}) = S(\langle 0.4, \ 0.7 \rangle) \quad = \tfrac{1}{2}(1 + 0.4 - 0.7)(\min(0.4, \ 0.7))^2$$
$$= \tfrac{1}{2}(0.7)(0.4)^2 = 0.35 \times 0.16 = 0.0056$$
$$S(c_{\widetilde{f}_{31}}) = S(\langle 0.7, \ 0.1 \rangle) \quad = \tfrac{1}{2}(1 + 0.7 - 0.1)(\min(0.7, \ 0.1))^2$$
$$= \tfrac{1}{2}(1.6)(0.1)^2 = 0.8 \times 0.01 = 0.008$$
$$S(c_{\widetilde{f}_{32}}) = S(\langle 0.2, \ 0.3 \rangle) \quad = \tfrac{1}{2}(1 + 0.2 - 0.3)(\min(0.2, \ 0.3))^2$$
$$= \tfrac{1}{2}(0.9)(0.2)^2 = 0.45 \times 0.04 = 0.018$$
$$S(c_{\widetilde{f}_{33}}) = S(\langle 0.5, \ 0.1 \rangle) \quad = \tfrac{1}{2}(1 + 0.5 - 0.1)(\min(0.5, \ 0.1))^2$$
$$= \tfrac{1}{2}(1.4)(0.1)^2 = 0.70 \times 0.01 = 0.007$$
$$S(c_{\widetilde{f}_{34}}) = S(\langle 0.6, \ 0.4 \rangle) \quad = \tfrac{1}{2}(1 + 0.6 - 0.4)(\min(0.6, \ 0.4))^2$$
$$= \tfrac{1}{2}(1.2)(0.4)^2 = 0.6 \times 0.16 = 0.096$$

Second objective costs:

$$S(c_{\widetilde{f}_{11}}) = 0.056, \quad S(c_{\widetilde{f}_{12}}) = 0.027$$
$$S(c_{\widetilde{f}_{13}}) = 0.1375, \quad S(c_{\widetilde{f}_{14}}) = 0.008$$
$$S(c_{\widetilde{f}_{21}}) = 0.036, \quad S(c_{\widetilde{f}_{22}}) = 0.048$$
$$S(c_{\widetilde{f}_{23}}) = 0.027, \quad S(c_{\widetilde{f}_{24}}) = 0.028$$
$$S(c_{\widetilde{f}_{31}}) = 0.056, \quad S(c_{\widetilde{f}_{32}}) = 0.064$$
$$S(c_{\widetilde{f}_{33}}) = 0.006, \quad S(c_{\widetilde{f}_{34}}) = 0.007$$

Third objective costs:

$$S(c_{\widetilde{f}_{11}}) = 0.018, \quad S(c_{\widetilde{f}_{12}}) = 0.162$$
$$S(c_{\widetilde{f}_{13}}) = 0.027, \quad S(c_{\widetilde{f}_{14}}) = 0.1375$$
$$S(c_{\widetilde{f}_{21}}) = 0.008, \quad S(c_{\widetilde{f}_{22}}) = 0.006$$
$$S(c_{\widetilde{f}_{23}}) = 0.162, \quad S(c_{\widetilde{f}_{24}}) = 0.056$$
$$S(c_{\widetilde{f}_{31}}) = 0.112, \quad S(c_{\widetilde{f}_{32}}) = 0.027$$
$$S(c_{\widetilde{f}_{33}}) = 0.014, \quad S(c_{\widetilde{f}_{34}}) = 0.008$$

Step 3: We obtain three individual transportation problems. Then, we solve these three transportation problems and obtain the basic feasible or optimal solutions for all problems.

For the first objective transportation problem:

$$F_1(w) = 0.014w_{11} + 0.016w_{12} + 0.027w_{13} + 0.1375w_{14} + 0.036w_{21} + 0.006w_{22}$$
$$+ 0.008w_{23} + 0.056w_{24} + 0.008w_{31} + 0.018w_{32} + 0.007w_{33} + 0.096w_{34}$$

subject to the constraints:

$$w_{11} + w_{12} + w_{13} + w_{14} \leq 0.096,$$
$$w_{21} + w_{22} + w_{23} + w_{24} \leq 0.036,$$
$$w_{31} + w_{32} + w_{33} + w_{34} \leq 0.048,$$
$$w_{11} + w_{21} + w_{31} \geq 0.014,$$
$$w_{12} + w_{22} + w_{32} \geq 0.056,$$
$$w_{13} + w_{23} + w_{33} \geq 0.096,$$
$$w_{14} + w_{24} + w_{34} \geq 0.014,$$
$$w_{ij} \geq 0, \sum_{i=1}^{m} a_i = \sum_{j=1}^{n} b_j$$

After solving this problem, we obtain the optimal solution as follows:

$$F_1 = 0.003090, w_{11} = 0.014, w_{12} = 0.056, w_{13} = 0.026, w_{14} = 0, w_{21} = 0,$$
$$w_{22} = 0, w_{23} = 0.022, w_{24} = 0.014, w_{31} = 0, w_{32} = 0, w_{33} = 0.048, w_{34} = 0.$$

For the second objective transportation problem:

$$F_2(w) = 0.0056w_{11} + 0.027w_{12} + 0.1375w_{13} + 0.008w_{14} + 0.036w_{21} + 0.048w_{22}$$
$$+ 0.027w_{23} + 0.028w_{24} + 0.056w_{31} + 0.064w_{32} + 0.006w_{33} + 0.007w_{34}$$

Subject to the constraints:

$$w_{11} + w_{12} + w_{13} + w_{14} \leq 0.096,$$
$$w_{21} + w_{22} + w_{23} + w_{24} \leq 0.036,$$
$$w_{31} + w_{32} + w_{33} + w_{34} \leq 0.048,$$
$$w_{11} + w_{21} + w_{31} \geq 0.014,$$
$$w_{12} + w_{22} + w_{32} \geq 0.056,$$
$$w_{13} + w_{23} + w_{33} \geq 0.096,$$
$$w_{14} + w_{24} + w_{34} \geq 0.014,$$
$$w_{ij} \geq 0$$
$$\sum_{i=1}^{m} a_i = \sum_{j=1}^{n} b_j .$$

After solving this problem, we obtain the optimal solution as follows:

$$F_2 = 0.005318, w_{11} = 0.014, w_{12} = 0.056, w_{13} = 0.012, w_{14} = 0.014, w_{21} = 0, w_{22} = 0,$$
$$w_{23} = 0.036, w_{24} = 0, w_{31} = 0, w_{32} = 0, w_{33} = 0.048, w_{34} = 0.$$

For the third objective transportation problem:

$$F_3(w) = 0.018w_{11} + 0.162w_{12} + 0.027w_{13} + 0.1375w_{14} + 0.008w_{21} + 0.006w_{22}$$
$$+ 0.162w_{23} + 0.056w_{24} + 0.112w_{31} + 0.027w_{32} + 0.014w_{33} + 0.008w_{34}$$

Subject to the constraints:

$$w_{11} + w_{12} + w_{13} + w_{14} \le 0.096,$$
$$w_{21} + w_{22} + w_{23} + w_{24} \le 0.036,$$
$$w_{31} + w_{32} + w_{33} + w_{34} \le 0.048,$$
$$w_{11} + w_{21} + w_{31} \ge 0.014,$$
$$w_{12} + w_{22} + w_{32} \ge 0.056,$$
$$w_{13} + w_{23} + w_{33} \ge 0.096,$$
$$w_{14} + w_{24} + w_{34} \ge 0.014,$$
$$w_{ij} \ge 0$$
$$\sum_{i=1}^{m} a_i = \sum_{j=1}^{n} b_j .$$

After solving this problem, we obtain the optimal solution as follows:

$F_3 = 0.00353$, $w_{11} = 0.014$, $w_{12} = 0$, $w_{13} = 0.082$, $w_{14} = 0$, $w_{21} = 0$, $w_{22} = 0.036$, $w_{23} = 0$, $w_{24} = 0$, $w_{31} = 0$, $w_{32} = 0.02$, $w_{33} = 0.014$, $w_{34} = 0$.

Step 4: After obtaining the solutions for all objectives individually, we obtain the pay off matrix such that:

|  | $\Sigma_1$ | $\Sigma_2$ | $\Sigma_3$ |
| --- | --- | --- | --- |
| $F_1$ | 0.003090 | 0.003965 | 0.004428 |
| $F_2$ | 0.007145 | 0.005318 | 0.015249 |
| $F_3$ | 0.015046 | 0.018077 | 0.003530 |

So, we can find the upper and lower bounds for all three objectives which are as follows:

$$L_1 = 0.003090, \ U_1 = 0.004428, \ d_1 = 0.001338.$$
$$L_2 = 0.005318, \ U_2 = 0.015249, \ d_2$$
$$L_3 = 0.003530, \ U_3 = 0.0180277, \ d_3 = 0.014547$$

.

Step 5: Now, model for the problem by using the proposed Fermatean fuzzy programming:

$$\text{Max } \gamma_1{}^3 - \gamma_2{}^3$$

where

$$\mu(F_k(w)){}^3 \ge \gamma_1{}^3, \ \forall \, k$$
$$\vartheta(F_k(w)){}^3 \le \gamma_2{}^3, \ \forall \, k$$

i.e., $\left(U_k - F_k(w)\right)^3 \ge d_k{}^3 \gamma_1{}^3, \ \forall \, k$

$$\implies \left(0.004428 - F_1\right)^3 \ge 0.00000000239534646 \gamma_1{}^3,$$
$$\implies \left(0.015249 - F_2\right)^3 \ge 0.000000979442501 \gamma_1{}^3,$$
$$\implies \left(0.018077 - F_3\right)^3 \ge 0.00000307836645 \gamma_1{}^3,$$

Again $\left(F_k(w) - L_k\right)^3 \le d_k{}^3 \gamma_2{}^3, \ \forall \, k$

$$\implies \left(F_1 - 0.003090\right)^3 \le 0.00000000239534646 \gamma_2{}^3,$$
$$\implies \left(F_2 - 0.005318\right)^3 \le 0.000000979442501 \gamma_2{}^3,$$
$$\implies \left(F_3 - 0.003530\right)^3 \le 0.00000307836645 \gamma_2{}^3,$$

with respect to the constraints,

$$w_{11} + w_{12} + \ldots\ldots + w_{1n} \leq a_1,$$
$$w_{21} + w_{22} + \ldots\ldots + w_{2n} \leq a_2,$$
$$.$$
$$.$$
$$.$$
$$w_{m1} + w_{m2} + \ldots\ldots + w_{mn} \leq a_m,$$
$$w_{11} + w_{21} + \ldots\ldots + w_{m1} \geq b_1,$$
$$w_{12} + w_{22} + \ldots\ldots + w_{m2} \geq b_2,$$
$$.$$
$$.$$
$$.$$
$$w_{1n} + w_{2n} + \ldots\ldots + w_{mn} \geq b_n$$

and $\sum_{i=1}^{m} a_i = \sum_{j=1}^{n} b_j$ , $w_{ij} \geq 0$, $0 \leq \gamma_1{}^3$, $\gamma_2{}^3 \leq 1$, $0 \leq \gamma_1{}^3 + \gamma_2{}^3 \leq 1$ and $\gamma_1{}^3 \geq \gamma_2{}^3$.

Then, by solving this model with the help of Lingo 19.0, we obtain the optimal solution such that:

$\gamma_1 = 1$, $\gamma_2 = 0.004699874$, $F_1 = 0.004400256$, $F_2 = 0.01528566$, $F_3 = 0.003598369$, $w_{11} = 0.014$, $w_{12} = 0$, $w_{13} = 0.082$, $w_{14} = 0$, $w_{21} = 0$, $w_{22} = 0.0350091$, $w_{23} = 0$, $w_{24} = 0.000990856$, $w_{31} = 0$, $w_{32} = 0$, $w_{33} = 0.014$, $w_{34} = 0.0130091$.

## 8. Conclusions

There are many approaches to convert fuzzy data into crisp data and many methods are introduced for the extension of fuzzy, i.e., for intuitionistic data, Pythagorean data, Fermatean data and other uncertain data. In this paper, we establish a new score function for the ranking of Fermatean Fuzzy numbers which helps to handle the Fermatean fuzzy uncertainty in a crisp environment. Then, we introduce a Fermatean Fuzzy Programming approach for Multi-objective Decision-Making Problems under uncertainty. Fermatean Fuzzy programming is non-linear programming for multi-objective problems which is an extension of Pythagorean Fuzzy Programming. With the proposed Fermatean Fuzzy programming approach, we built a model of a MOTP and solved a numerical illustration of a MOTP in the Fermatean Fuzzy environment. We found that our proposed approach is fruitful in finding a compromise optimal solution for Multi-objective Decision-Making Problems. Therefore, we can say that our proposed methodology is an alternate way for solving multi-objective decision-making problems in the Fermatean fuzzy environment and we can also use the proposed Fermatean fuzzy programming approach to solve multi-objective decision-making problems in any other fuzzy environment. In the end, for future perspectives, we will enhance the technique of our type-2 fuzzy logic and develop a model to handle many engineering and medical areas.

**Author Contributions:** Conceptualization, M.K.S. and K.; methodology, M.K.S. and K.; software, A.D., F.E.L.M. and A.G.H.; validation, A.N.; formal analysis, A.N. and M.K.S.; investigation, M.K.S., K. and A.D.; resources, H.G.R., V.T.H., A.D., F.E.L.M. and A.G.H.; data curation, M.K.S. and K.; writing—original draft preparation, M.K.S.; writing—review and editing, A.N., H.G.R., V.T.H., A.D. and A.G.H.; visualization, H.G.R. and A.G.H.; project administration, A.D.; funding acquisition, H.G.R., F.E.L.M. and A.G.H. All authors have read and agreed to the published version of the manuscript.

**Funding:** This research received no external funding.

**Acknowledgments:** This work is supported by Universidad Autonoma de Zacatecas, Mexico and CONACyT, Mexico.

**Conflicts of Interest:** The authors declare no conflict of interest.

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
