# Peer review of "Fermatean Fuzzy Programming with New Score Function: A New Methodology to Multi-Objective Transportation Problems"

_electronics, doi:10.3390/electronics12020277_

Round 1

Reviewer 1 Report

High similarity index (27%), suggest to reduce to below 18%. 

Problem statements and research gap amongst these techniques need to be highlighted. 

Please look into the writing style. Suggest for editing services. 

This paper utilize Fermatean fuzzy technique to deal with multi-objective problems and able to convert fuzzy data into crisp data. The overall scope of this paper is considered narrow, suggest to perform a complete simulation works such that some outcomes results can be generated. Not limited to data conversion processes. 

Author Response

First Reviewer Comments

Authors reply

Moderate English changes required

The English of the work has been improved in the revised manuscript.

Does the introduction provide sufficient background and include all relevant references?

As per the suggestion of the reviewer the introduction has been improved to provide sufficient background and include all relevant references.

Is the research design appropriate?

The research design has been improved in the revised manuscript.

Are the results clearly presented?

The results of this work are clearly presented in the revised manuscript.

Are the conclusions supported by the results?

The conclusion has been enhanced to support the results of this work in the revised manuscript.

High similarity index (27%), suggest to reduce to below 18%. 

As per the suggestion of the reviewer the similarity has been reduced in the revised manuscript.

Problem statements and research gap amongst these techniques need to be highlighted. 

The statement of the problem with research gap has been added in the revised manuscript.

Please look into the writing style. Suggest for editing services. 

The writing style has been improved in the revised manuscript.

Reviewer 2 Report

The paper aims to propose an alternate fuzzy programming approach to solve the multi-objective transportation problem.

The introduction should be improved, e.g. add the author's name before a citation at the beginning of some sentences ([4] introduced decision-making problems under such environment in which 62 goal or constraints are not defined precisely, etc.). In addition, references are not cited according to their appearance in the text. Authors do not comment the research gap in the reviewed studies.

The research methodology is not well described. Sections 2, 3 and 4 are very short and difficult to understand.

I strongly recommend authors to describe their methodology presented in Section 6 and Numerical calculations in Section 7 in more details.

In the conclusion, authors do not comment the limitations of their study and their plans for future work in the field

Author Response

Second Reviewer Comments

Authors reply

English language and style are fine/minor spell check required.

The English language of the manuscript has been improved in the revised manuscript.

Does the introduction provide sufficient background and include all relevant references?

As per the suggestion of the reviewer the introduction has been improved to provide sufficient background and include all relevant references.

Are all the cited references relevant to the research?

As per the suggestion all the references have been improved accordingly.

Is the research design appropriate?

The research design has been upgraded in the revised manuscript.

Are the methods adequately described?

The method is adequately described in the revised manuscript.

Are the results clearly presented?

The results of this work are clearly presented in the revised manuscript.

Are the conclusions supported by the results?

The conclusion has been enhanced to support the results of this work in the revised manuscript.

The introduction should be improved, e.g. add the author's name before a citation at the beginning of some sentences ([4] introduced decision-making problems under such environment in which 62 goal or constraints are not defined precisely, etc.). In addition, references are not cited according to their appearance in the text. Authors do not comment the research gap in the reviewed studies.

The introduction is improved now. The author’s name has been added before the citation.  The references are cited according to their appearance in the text. The research gap is added in the revised manuscript.

The research methodology is not well described. Sections 2, 3 and 4 are very short and difficult to understand.

The Sections 2, 3 and 4 are enhanced so that one can understand easily.

I strongly recommend authors to describe their methodology presented in Section 6 and Numerical calculations in Section 7 in more details.

As per the suggestion of the reviewer, the methodology given in section 6 and 7 has been improved in the revised manuscript.

In the conclusion, authors do not comment the limitations of their study and their plans for future work in the field

In the conclusion, the limitations and future plans has been added to support the credibility of this work in the revised manuscript.

Round 2

Reviewer 1 Report

Much effort and improvement shown in the revised manuscript. Hence, i am agreed to accept this paper to be published in "Electronics".

Reviewer 2 Report

I am satisfied with the changes made and I think the paper can be published